# Identification of a Novel Idiopathic Epilepsy Risk Locus and a Variant in the *CCDC85A* Gene in the Dutch Partridge Dog

**DOI:** 10.3390/ani13050810

**Published:** 2023-02-23

**Authors:** Evy Beckers, Sofie F. M. Bhatti, Mario Van Poucke, Ingeborgh Polis, Frédéric Farnir, Filip Van Nieuwerburgh, Paul Mandigers, Luc Van Ham, Luc Peelman, Bart J. G. Broeckx

**Affiliations:** 1Department of Veterinary and Biosciences, Faculty of Veterinary Medicine, Ghent University, 9820 Ghent, Belgium; 2Small Animal Department, Faculty of Veterinary Medicine, Ghent University, 9820 Ghent, Belgium; 3Biostatistics and Bioinformatics Applied to Veterinary Science, FARAH Research Centre, Faculty of Veterinary Medicine, University of Liège, 4000 Liège, Belgium; 4Department of Pharmaceutics, Faculty of Pharmaceutical Sciences, Ghent University, 9000 Ghent, Belgium; 5Department of Clinical Sciences, Faculty of Veterinary Medicine, Utrecht University, 3584 CL Utrecht, The Netherlands

**Keywords:** canine, genome-wide association study, whole-exome sequencing, *GRIK2*

## Abstract

**Simple Summary:**

A significant percentage of Dutch partridge dogs suffers from epileptic seizures. Previous studies indicated that this is likely the result of genetic idiopathic epilepsy, but to date, little to no knowledge exists on the genetic cause. This study aimed to identify (a) causal variant(s) and/or risk loci associated with the disease. A risk locus for idiopathic epilepsy was described for the first time on chromosome 12. Furthermore, a variant in the *CCDC85A* gene was found to increase the risk of disease in homozygous variant dogs. Further research should be conducted to determine whether the chromosome 12 risk locus and *CCDC85A* variant can be used in breeding decisions.

**Abstract:**

(1) Idiopathic epilepsy (IE) is thought to have a genetic cause in several dog breeds. However, only two causal variants have been identified to date, and few risk loci are known. No genetic studies have been conducted on IE in the Dutch partridge dog (DPD), and little has been reported on the epileptic phenotype in this breed. (2) Owner-filled questionnaires and diagnostic investigations were used to characterize IE in the DPD. A genome-wide association study (GWAS) involving 16 cases and 43 controls was performed, followed by sequencing of the coding sequence and splice site regions of a candidate gene within the associated region. Subsequent whole-exome sequencing (WES) of one family (including one IE-affected dog, both parents, and an IE-free sibling) was performed. (3) IE in the DPD has a broad range in terms of age at onset, frequency, and duration of epileptic seizures. Most dogs showed focal epileptic seizures evolving into generalized seizures. A new risk locus on chromosome 12 (BICF2G630119560; p_raw_ = 4.4 × 10^−7^; p_adj_ = 0.043) was identified through GWAS. Sequencing of the *GRIK2* candidate gene revealed no variants of interest. No WES variants were located within the associated GWAS region. However, a variant in *CCDC85A* (chromosome 10; XM_038680630.1: c.689C > T) was discovered, and dogs homozygous for the variant (T/T) had an increased risk of developing IE (OR: 6.0; 95% CI: 1.6–22.6). This variant was identified as likely pathogenic according to ACMG guidelines. (4) Further research is necessary before the risk locus or *CCDC85A* variant can be used for breeding decisions.

## 1. Introduction

Epilepsy is one of the most common neurological diseases in dogs, with an estimated prevalence of 0.62–0.82% in the general dog population [1,2,3]. It is caused by sudden, excessive neuronal activity in the brain, leading to epileptic seizures characterized by motor, autonomic, and/or behavioral signs. According to the current recommendations of the International Veterinary Epilepsy Task Force (IVETF), canine epilepsy can be classified by etiology as reactive, structural, or idiopathic [4]. Idiopathic epilepsy (IE) can be subclassified into IE with a genetic cause, IE with a suspected genetic cause, and IE of unknown cause. A high breed prevalence is indicative of a genetic cause and has been reported in multiple predisposed breeds such as the Labrador retriever, petit basset griffon vendees, Belgian shepherd, Irish wolfhound, pug, boxer, basset hound, border terrier, and border collie [1,5,6,7,8].

Although a genetic cause is suspected in many breeds, only three causal variants of epilepsy have been identified to date: a dodecamer repeat in the *NHLRC1* gene that causes progressive myoclonic epilepsy (Lafora disease) first reported in the wirehaired dachshund (OMIA 000690-9615) [9], a nonsense variant in *LGI2* that causes benign familial juvenile epilepsy in the Lagotto Romagnolo (OMIA 001596-9615) [10], and a deletion in the *DIRAS1* gene that is the causal variant for juvenile myoclonic epilepsy with photosensitivity in the Rhodesian ridgeback (OMIA 002095-9615) [11]. Whereas the two latter diseases are classified as idiopathic epilepsy, Lafora disease is not since it has a structurally metabolic etiology [12]. A deletion in *PITRM1* causes a neurodegenerative syndrome in the Parson Russel terrier, with severe epileptic seizures as the main clinical sign (OMIA 002324-9615) [13]. As with Lafora disease, these epileptic seizures have a metabolic etiology.

One of the many breeds without an identified causal variant for IE is the Dutch partridge dog (DPD), a Dutch hunting breed. Epilepsy has been reported in this breed since 1986, and several findings suggest a genetic component. A study published in 1986 demonstrated a familial predisposition to epilepsy [14]. Secondly, IE in the DPD has a high prevalence; the 1986 study estimated a prevalence of at least 1.4% [14], and another study reported an incidence of 2.2% between 2006 and 2011 [15]. Prevalences exceeding 2% support a genetic basis according to the IVETF [16]. Finally, the study published in 1986 found a narrow sense heritability (h²) between 0.33 and 0.47 for IE [14].

No genomic studies have been conducted to date on IE in the DPD. Therefore, this study aimed to identify IE risk loci and putative causal variants in this breed. First, IE was phenotypically characterized through the use of owner-filled questionnaires and diagnostic investigations performed by a veterinarian. Next, two parallel genomic methods were used. First, a traditional genome-wide association study (GWAS) was performed to identify a candidate region associated with IE in the DPD. Secondly, a heuristic approach was applied, using whole-exome sequencing (WES) of a single family to identify family-specific variants.

## 2. Materials and Methods

### 2.1. Study Cohort

DPDs with and without epilepsy were recruited through a call to Belgian and Dutch breeder organizations and a team of independent breeders. Blood was collected in EDTA-laced tubes for all 292 dogs (between November 2015 and February 2018) and stored at −20 °C. Signed informed consents were collected from the dogs’ owners, and the sample collection was approved by the Ethics Committee of the Faculties of Veterinary Medicine and Bioscience Engineering at Ghent University (EC2017-86). The approval date is 5 February 2018.

#### 2.1.1. Healthy Controls

Dogs were included as controls were at least 6 years old, healthy, and had never shown signs of epileptic seizures during their life.

#### 2.1.2. Epileptic Dogs

To assess whether dogs had IE, questionnaires were sent to the owners of 46 epileptic dogs (translated questionnaire, Appendix A), and telephone interviews were conducted to clarify or supplement answers provided in the questionnaires. The questionnaires gathered information about the dogs’ signalment; characteristics of the ictal, pre-, and post-ictal phases; (epileptic) seizure frequency, duration, and severity; age of onset of (epileptic) seizures; possible trigger factors; diagnostic investigations performed; and antiseizure medication (ASM) that was being administered. When available, videos of the seizures were evaluated by a trained neurologist.

Dogs were considered to suffer from IE (hereafter referred to as cases) when no evidence of a reactive seizure or structural epilepsy was identified and when history, as well as clinical and neurological signs, agreed with 6 inclusion criteria: (1) the presence of muscle contractions (tonic and/or clinical, orofacial, or any other body part), (2) age of onset between 6 months and 6 years, (3) at least 2 autonomous symptoms (salivation, urination, defecation, or loss of consciousness), (4) presence of a post-ictal phase, (5) absence of neurological symptoms in between epileptic seizures, and (6) absence of head trauma.

#### 2.1.3. Statistical Analysis

A Fisher exact test was performed to look for a sex predisposition.

### 2.2. Genome-Wide Association Analysis

Samples of 200–500 µL EDTA blood of 16 cases and 43 controls (i.e., the samples available at the time of the GWAS analysis) were sent to the GIGA-Genomics platform, University of Liège. The samples were genotyped for 172115 SNVs using an Illumina CanineHD BeadChip according to the manufacturer’s instructions (Illumina, Inc., San Diego, CA, USA).

Raw genotype data were subjected to a 4-step quality control process using PLINK v1.9 [17]. (1) SNVs with quality scores <70% were removed (based on the GenTrainScore in GenomeStudio); (2) SNVs and individuals with levels of missingness exceeding 2% were consecutively filtered out (--geno and --mind filters in PLINK); (3) SNVs with a minor allele frequency below 5% were removed (--maf filter in PLINK); (4) SNVs with a significant difference in missingness between cases and controls were removed (--test-missing in PLINK).

GWAS analysis was performed with the remaining 99424 SNVs and samples using Fisher’s exact testing for allelic association in PLINK. To evaluate population stratification, a QQ plot was created in R v3.6.3 [18] using the qqman package [19]. To correct for the observed population stratification, a principal component analysis (PCA) was performed with a subset of 33779 genotyped SNVs. In order to retain the most informative SNVs, this subset only contained SNVs with squared allele count correlations (r²) < 0.5 within a 200 kb window using a sliding window approach (with 5 bp steps) (--indep-pairwise 200 kb 5 0.5 in PLINK). The first principal component was included in a univariate linear mixed model association test using GEMMA software [20].

Bonferroni-adjusted *p*-values (i.e., *p*-value multiplied by the number of tests [21]) were used to identify genome- and chromosome-wide significant SNVs. Odds ratios (ORs, see Appendix A, alleles for calculations) and 95% confidence intervals (95% CIs) were calculated for significant SNVs. Manhattan plots were created using the qqman package [19].

### 2.3. GWAS Candidate Gene

The *GRIK2* gene is an interesting candidate gene within a GWAS-established region; its coding sequence (including splice sites) was further examined using Sanger sequencing in six cases and one control sample. Genomic DNA was extracted from all samples as previously described [22]. Primers flanking the coding exons of the gene were designed with NCBI Primer-BLAST based on the latest canine reference assembly, ROS_Cfam_1.0 (Acc. No. GCF_014441545.1; *GRIK2* location: NC_051816.1: g.59766364_60827294), taking into account secondary structures (mFold [23]), known SNVs (employing SNV handling in Primer-BLAST [24]), and repeat sequences (repeat filter set to automatic)). Subsequently, a PCR reaction was performed for each primer pair, after which 2 µL of PCR product was loaded on agarose gel electrophoresis, while the remaining 8 µL was used to perform Sanger sequencing (using both primers), as previously described [25]. *GRIK2* exon information can be viewed in Appendix A, primer sequences and amplicon lengths are presented in Appendix A, and details of PCR/sequencing mixes and programs can be viewed in Appendix A. Five known SNVs and one new variant were detected in the cases.

### 2.4. Whole-Exome Sequencing

While GWAS was used to detect IE-associated regions and possible common genes, we performed an additional WES on a nuclear family to screen for putative family-specific variants as a parallel method. This family included a case, both seizure-free parents, and a seizure-free sibling. In humans, generalized epilepsies have displayed a Mendelian inheritance pattern in some cases, and several rare, family-specific deleterious variants have been identified using whole-exome sequencing [26].

Blood was extracted from all four samples using a DNeasy blood and tissue kit (Qiagen, Hilden, Germany) according to the spin column protocol (the elution step was performed twice with 50 µL molecular-grade water instead of 200 µL buffer EA), and WES was performed as described by Broeckx et al. [27]. Reads were aligned to the CanFam3.1 reference genome using BWA v0.7.17 [28], and duplicate reads were marked with Picard v2.21.6. Variants were called using GATK’s HaplotypeCaller in GVCF mode according to the GATK Best Practices (GATK v4.1.2.0) [29]. Subsequently, all 403562 variants were filtered with GATK VariantFiltration according to the “hard filtering” described in the GATK Best Practices. Using custom R scripts (R v3.6.3) [18], the remaining variants were only retained if they were absent in the variant database of the Animal Genetics Lab (Sequence Read Archive (SRA) accession number PRJNA891496) and followed an autosomal recessive mode of inheritance (which is the inheritance pattern that was detected for this family). Based on annotation according to the online variant effect predictor (VEP) tool [30], non-coding and synonymous variants were removed.

For the 80 remaining variants, the function of the genes in which they were located was inspected for a potential link to epilepsy. Using the R script provided by Broeckx et al. [31], a threshold value (Tv) was calculated to filter out variants with a minor allele frequency (MAF) > Tv in the European Variation Archive (EVA) database [32]. The disease prevalence (Pd) was set to 0.8%, coverage was set to 0.95, data sampling was set to “a”, and an autosomal recessive inheritance pattern was presumed. The database population size was altered to correspond to the size for each variant. A total of 5 variants were retained.

### 2.5. Variant Analysis and ACMG Classification

Candidate variants identified in the *GRIK2* gene and by WES were further examined for their pathogenicity. The effects at the protein level of the newly discovered variants in the *GRIK2* gene (absent in the reference sequence (ROS_Cfam_1.0) and the control sample) and of the remaining WES variants were evaluated with the PROVEAN online prediction tool [33], PolyPhen-2 [34], and MutPred2 [35] (in case of indels, MutPred-Indel [36]). Only variants for which at least one tool indicated a deleterious effect were considered for further investigation.

The remaining potentially deleterious variants (in *ENAH*, *CCDC85A*, *VPS54*, *SPAST*, and *BRINP3*) were subsequently genotyped in all 18 cases and 18 controls (individuals other than the family members of the WES case). The *ENAH* variant was genotyped using PCR followed by gel electrophoresis on a 3% agarose gel, while the other variants were genotyped using Sanger sequencing, as described above for *GRIK2* sequencing. Primer sequences, amplicon lengths, and the primers used for sequencing are shown in Appendix A. The variant allele frequencies were compared between cases and controls, and ORs and 95% CIs were calculated for each variant (calculations according to Appendix A, alleles). Variants for which an OR above 5 was found were further examined, in agreement with the standards and guidelines for the interpretation of sequence variants by the American College of Medical Genetics and Genomics (ACMG) [37].

The two remaining variants in the *CCDC85A* and *SPAST* genes were genotyped in all dogs included in the study, and the variant allele frequency (Vt%) was calculated for the entire population, as well as for the selected cases and controls. The number of homozygous variant (Vt/Vt) dogs was compared to the number of homozygous wild-type and heterozygous dogs (Wt/Wt + Wt/Vt) between selected cases and controls with a Fisher exact test, and the OR (calculations according to Appendix A, genotypes) and 95% CI was calculated again. The presence of a homologous variant was examined in human variant databases. Variant classification was performed using the aforementioned ACMG guidelines.

## 3. Results

### 3.1. Study Cohort

A total of 292 DPDs entered this study, including 45 dogs displaying epileptic seizures according to their owner and 247 healthy dogs. Most dogs were from the Netherlands (201/292 = 69%) or Belgium (84/292 = 29%), and some were from Germany and Denmark (4/292 and 3/292, respectively; both = 1%). The sample represents dogs from many different families, with some closely related individuals. For 52 litters, two or more full siblings were available, and for 18 of the dogs displaying epileptic seizures, a normal littermate was available. The sire was available for 25 litters (14 sires in total), and the dam was available for 32 litters (22 dams in total). In total, 135 dogs were male, and 149 were female (sex was unknown for 8 dogs).

#### 3.1.1. Healthy Controls

In total, 100 of the 247 seizure-free dogs were included as healthy controls, while the 147 remaining healthy dogs did not meet the age requirement. Of these 100 dogs, 82 were over 10 years old, 9 were between 8 and 10 years old, and 9 were between 6 and 8 years old. Several of the control dogs were included in the GWAS analysis (see below).

#### 3.1.2. Diagnostic Examination

A blood examination was performed for 24 epileptic dogs, and results for 10 dogs were available for inspection by the authors. The minimum database blood test (as recommended by the IVETF [16]) was performed for seven of these dogs, while complete blood serum biochemistry was missing for the three remaining dogs. Urinalysis was performed on seven dogs and included sediment cytology, protein, glucose, and pH for five dogs. Specific gravity was not determined in one dog, and for another dog, no details were available. No significant abnormalities were detected in the blood or urine tests. Thirteen dogs were examined neurologically by their veterinarian, with no abnormalities detected, except in two dogs that were examined shortly after an epileptic seizure. A magnetic resonance imaging (MRI) scan of the brain and cerebrospinal fluid (CSF) analysis was available for one dog, and a CT of the brain was available for another dog; neither showed significant abnormalities. In total, five dogs were diagnosed with IE at a tier I/II confidence level as described by the IVETF [16].

#### 3.1.3. Epileptic Cases

The questionnaire was completed for 42 out of 45 dogs suffering from epilepsy, and additional video material was available for 7 of them. All videos were analyzed by a veterinary neurologist (Dr. Sofie F.M. Bhatti), and one was included as a Appendix A. Two dogs only displayed one epileptic seizure and were excluded from further analysis. Seven more dogs were excluded because the age of onset of seizures was too low (2 months for one dog) or too high (six dogs; age of onset up to 10 years old). Fourteen dogs were not included because they did not display at least two autonomous symptoms, and another dog was excluded because it had no muscle contractions. The remaining 18 dogs met all inclusion criteria and were included as cases in this study.

No significant sex predispositions were found (*p* > 0.05). The mean age of onset of epileptic seizures was 3.5 years old (s.d. = 1.4; ranging from 6 months to 5 years old). Two-thirds of the dogs had focal epileptic seizures evolving into generalized epileptic seizures, and the rest had generalized tonic–clonic epileptic seizures. Epileptic seizures could be predicted by the owner in 67% of the cases. A wide range of seizure frequency was reported by the owners (from more than once per week to less than once per year), and 39% of the dogs showed cluster seizures (two or more epileptic seizures within 24 h). Epileptic seizures never lasted longer than 15 min, and most lasted between 2 and 5 min (50%) or between 1 and 2 min (28%). Status epilepticus (an epileptic seizure longer than 5 min) was reported in three dogs (17%). Almost all dogs (83%) received ASM, and 80% of this group received a combination of multiple drugs. In 11 dogs (73%), the owners noticed an improvement in seizure frequency, severity, and/or duration after starting medical treatment, with seizure freedom in 2 dogs. A complete overview of seizure characteristics and medical treatment can be found in Table 1.

### 3.2. Genome-Wide Association Analysis

In total, 16 of the 18 cases and 43 of the 100 healthy controls (all more than 10 years of age) were available at the time of the GWAS analysis. After quality control filtering, all dogs and 99424 variants remained. Allelic association testing using a Fisher exact test with Bonferroni correction revealed one genome-wide significantly associated SNV (BICF2G630119560; p_raw_ = 2.6 × 10^−7^; p_adj_ = 0.026) on chromosome 12. To delimit a region on this chromosome, chromosome-wide significance was inspected among the SNVs on chromosome 12 using Bonferroni correction, revealing nine other SNVs that reached chromosome-wide significance using this method. Manhattan and QQ plots are shown in Appendix A, respectively.

As the QQ plot with the *p*-values of the Fisher exact test did show a clear deviation (Appendix A), 33779 variants were used to perform a PCA (Appendix A). Association testing was performed with a univariate linear mixed model, including the first principle component as a covariate. After Bonferroni correction, the same SNV remained significant genome-wide (BICF2G630119560; p_raw_ = 4.4 × 10^−7^; p_adj_ = 0.043). This time, only two other SNVs reached chromosome-wide significance on chromosome 12, identifying a 0.5 Mb candidate region. An overview of the associated SNVs on chromosome 12 with ORs and 95% CIs can be found in Appendix A. Manhattan plots are displayed in Figure 1a,b, and the QQ plot is shown in Appendix A.

The 0.5 Mb associated region only includes one gene on the Ensembl 104 annotation (Figure 1C), *GRIK2* (NCBI Gene ID 481938), which is an interesting candidate gene. Therefore, the coding and splice site regions of *GRIK2* were sequenced in six cases and compared to the latest reference genome (ROS_Cfam_1.0), as well as to a control dog. Five known SNVs (rs851724710, rs8745274, rs8745273, rs850859789, and rs9151294) and one new variant were identified in the cases. Of the known SNVs, three are synonymous variants, and two are intronic variants. For both intronic variants, a high variant allele frequency is reported in the EVA database (47% and 32% for rs8745274 and rs850859789, respectively). Therefore, none of the known SNVs were investigated further. The new variant was absent in the control sample (NC_051816.1 (XM_038684247.1): c.589C > T (p.(Leu197Phe))) and therefore further examined (see below, variant analysis and ACMG classification) (Table 2).

### 3.3. Whole-Exome Sequencing

After WES of one case, both seizure-free parents, and a seizure-free sibling, 403562 variants (SNVs and indels) were called. Following hard filtering and only retaining variants that were not present in the variant database of the Animal Genetics Lab (Ghent University, SRA PRJNA891496) and that followed an autosomal recessive mode of inheritance, 2592 variants remained. Of these, 80 were protein-coding and not synonymous. None of these variants was located within the GWAS candidate region on chromosome 12.

For the variants located in genes with a function potentially linked to epilepsy, the maximum expected allelic frequency in the variant database (i.e., “the Tv threshold”) was calculated to be 10% (lowest Tv = 10.1% and highest Tv = 10.3%). Five variants with an MAF below this threshold in the EVA database were retained in this manner (Table 2).

### 3.4. Variant Analysis and ACMG Classification

Three prediction tools were used to estimate the effect of the *GRIK2* variant and the five remaining WES variants on the produced protein. Only the *GRIK2* variant was not predicted as deleterious by any of the prediction tools (Table 3).

The 5 WES variants were genotyped in all 18 cases and in 18 controls (Appendix A), and a significant difference in variant allele frequency was found for the *CCDC85A* (NCBI gene ID: 481377; OR: 8.5; 95% CI: 1.7–41.5) and *SPAST* (NCBI gene ID: 608582; OR: 7.1; 95% CI: 1.2–42.9) variants. According to the ACMG standards and guidelines, we endeavored to classify these two variants as (likely) pathogenic, (likely) benign, or of uncertain significance [37]. Subsequently, all 292 samples were genotyped for both variants. Unfortunately, it was not possible to gather enough family material to investigate variant segregation in affected family members in favor of or against pathogenicity (PP1 or BS4, respectively) for either variant in this study. Genotyping results for the case group, control group, and total population, as well as Vt%, are displayed in Table 3.

A significant difference (*p* = 0.012) was found for the *CCDC85A* variant when comparing the number of Vt/Vt dogs to the number of dogs that were either Wt/Wt or Wt/Vt between cases and controls, with an OR of 6.0 (95% CI: 1.6–22.6), providing strong evidence of pathogenicity (PS4). Additional moderate evidence (PM2) was provided by the very low occurrence of the variant in population databases. The Vt% is 0.5% in dogs in the EVA database. Since the same *CCDC85A* missense variant is described for humans in the dbSNP database (rs1052182749), its allele frequency was verified in human populations. The Vt% of the homologous variant in humans (rs1052182749) is much lower than in dogs (0.0016% in gnomAD exomes, 0.0032% in gnomAD genomes, and 0.0026 in TopMed). Computational evidence could not be used as supporting evidence (PP3/BP4) in the variant classification, as two algorithms predicted a deleterious effect at the protein level, whereas one other predicted a neutral effect (see Table 2). Little is also known about the potential role of *CCDC85A* in epilepsy pathogenesis, and few functional studies have been conducted (PS3/BS3). In summary, two relevant criteria were fulfilled (one strong (PS4) and one moderate (PM2)), which is sufficient to classify this variant as likely pathogenic (Table 4).

No strong evidence of pathogenicity was found for the *SPAST* variant. Although a significant difference (*p* = 0.049) was found when comparing the allele frequencies between cases and controls, with an OR of 2.6 (95% CI: 1.0–6.4), the OR was below 5, which does not fulfill the PS4 criterion. A homologous variant in *SPAST* is also described in humans in the dbSNP database (rs1268625226), and low MAF values are reported in multiple studies (0.0007% in gnomAD genomes, 0.0008% in TopMed, and not detected in NCBI ALFA). These very low frequencies in human databases and the relatively low frequency in the EVA database (Vt% = 1.6%) provided moderate evidence for pathogenicity (PM2). As with *CCDC85A*, little is known about *SPAST*’s role in epilepsy pathogenesis, and the PS3 criterion is not fulfilled. Supportive evidence is provided by the three algorithms, predicting a deleterious effect at the protein level (PP3), and by missense variants in *SPAST* (including in the microtubule interacting and transport (MIT) domain), which is a common mechanism for disease (PP2) [38]. Since only one moderate (PM2) and two supportive criteria are met (PP2 and PP3), this variant is of uncertain significance according to the ACMG guidelines.

## 4. Discussion

This study faced the standard complexity associated with epilepsy studies. The intricate inheritance of epilepsy is well known in human genetics, and the genetic understanding of the disease is limited [26]. Genetic epilepsies range from simple monogenic to complex polygenic with considerable interaction of environmental factors, leading to a broad phenotypic spectrum [39]. Although dogs generally provide a good model for GWAS because of the extensive occurrence of linkage disequilibrium (up to 100 times more extensive than in humans) [40], genetic epilepsy studies deal with similar problems as seen in human studies. While a genetic etiology is suspected in many dog breeds, very few causal variants and risk loci have been identified for IE to date, probably because of the genetic complexity of this disease [41]. Moreover, correct phenotyping of IE, the crucial step in any genetic study [42] represents another hurdle. The diagnosis is exclusion-based, and many diagnostic tests are required to differentiate epilepsy from other similar neurological diseases and to rule out metabolic or structural causes of epilepsy. The aforementioned difficulties make a final diagnosis of IE with a suspected genetic cause quite challenging.

The aim of this study was to find a causal variant for IE in the DPD, a breed in which the prevalence of IE is high, with evidence pointing towards a genetic cause, and to phenotypically characterize the disease. To attempt correct IE phenotyping following the IVETF guidelines, information was gathered regarding phenotypical characteristics and diagnostics. While phenotyping based on an owner-filled questionnaire is arguably a limitation of this study, the questionnaire was very extensive to make the best possible estimate of the patients’ phenotype. Moreover, any diagnostic information determined by the dogs’ veterinarians was collected, and video material was collected for seven dogs. This led to sufficient information to diagnose five dogs with IE at a tier I/II confidence level [16]. Strict inclusion criteria were used in this study, limiting the erroneous inclusion of dogs as cases.

As a result of the questionnaires, the phenotype of IE in DPDs was described. Notably, several dogs were excluded because the age of onset was too low or too old. One dog even started showing epileptic seizures for the first time at 10 years of age. The fact that epilepsy in this breed can develop at a later age than usual has been reported before [15]. While a higher age limit to include dogs as cases could have been used in this study, we decided to use a more stringent age of onset of between 6 months and 6 years old [16], as this improves the likelihood of correct phenotyping, and a uniform phenotype is of great importance in genetic studies [42]. In the epileptic dog population, status epilepticus was observed in 21% of the dogs. Cluster seizures occurred in 23% of dogs, which greatly exceeds the frequency of <10% reported by Mandigers (2017) [15]. However, that study reported the occurrence of three or more seizures within one time unit, while an occurrence of two or more was used here, as proposed by the IVETF [4], possibly explaining this difference. Very similar to what was reported in that study, seizures occur more often in males than females (58% male vs. 42% female). When only looking at the selected cases, the occurrence is slightly higher in females (44% male vs. 56% female). However, contrary to the great Swiss mountain dog (GSMD), in which males are more likely to be affected than females [43], no significant sex predisposition was found for the DPD in the current study, nor the study by Mandigers [15].

About two-thirds of the epileptic cases in this study displayed generalized seizures evolving from focal seizures, similar to what was described for the GSMD [43], in contrast to the border collie, in the vast majority of which have generalized seizures and only 4% of which display a combination of both seizure types [44]. Similar to what is reported in the border collie and GSMD [43,44], the majority of the dogs (83%) received medical treatment. The number of DPDs receiving single and combination drug therapy is comparable to the border collie, whereas the opposite is true for the GSMD.

The lack of identified causal variants and susceptibility loci point to complex inheritance patterns of canine IE, as is also seen in human IE [45]. To date, only two studies have identified a causal variant for IE in dogs [10,11], and one study identified a risk locus in the *ADAM23* gene for the Belgian shepherd [46]. Later, a common 28 kb risk haplotype in the same gene was established in the Belgian shepherd, as well as the schipperke, Finnish spitz, and beagle [47]. The same risk haplotype was subsequently found to be associated with IE in four other breeds (Australian shepherd, Kromfohrländer, Labrador retriever, and whippet), proposing *ADAM23* as a common risk gene for epilepsy with low penetrance [48]. A recent study aimed to identify additional risk loci in the Belgian shepherd. The authors found a significant association between IE and a locus on chromosome 14, while a suggestive association was found adjacent to the previously described *ADAM23* locus [49]. Yet another study in the Belgian shepherd again validated the chromosome 14 and 37 risk haplotypes and identified an insertion in the *RAPGEF5* gene adjacent to the chromosome 14 haplotype [50]. While some studies have been successful, the majority have not.

Here, a new risk locus was identified on chromosome 12. A GWAS analysis of 16 cases, 43 controls, and 99424 SNVs using a univariate linear mixed model test and Bonferroni correction revealed one SNV that was significantly associated with IE in the DPD genome-wide (BICF2G630119560; praw = 4.4 × 10^−7^; padj = 0.043). An OR of 10.4 was calculated for the associated allele (95% CI: 4.1–26.5). To the best of our knowledge, this is the first time that this locus has been identified as a susceptibility locus for IE in dogs. A 0.5 Mb candidate region was delineated through the location of two additional chromosome-wide associated SNVs on the same chromosome. Inspection of that region identified an interesting candidate gene, *GRIK2*, located 0.3 Mb upstream of BICF2G630119560.

The *GRIK2* gene encodes the kainite receptor (KAR) subunit GluK2 (previously known as GluR6). KARs are ionotropic glutamate receptors widely expressed in the central nervous system [51] and are tetramers that can be formed from subunits GluK1-5 [52]. They are key players in glutaminergic synaptic transmission in the hippocampus and probably other brain structures. KARs can be activated by glutamate, after which gamma-aminobutyric acid (GABA) release is depressed and the excitability of neuronal cells is increased [51,52,53]. Epilepsies have been linked to KARs, including GluK2 [54], and Grik2 knockout mice have reduced susceptibility to seizures induced by kainate, a high-affinity KAR agonist [55]. Furthermore, overexpression of the Grik2 kainate receptor in the hippocampus induces seizures in rats [56]. In humans, post-translational modifications of *GRIK2* were shown to cause/influence epileptic phenotypes, as was seen in mesial and lateral temporal lobe epilepsy patients [52,57,58]. Furthermore, multiple variants were linked to epilepsy susceptibility in a Chinese population and a cohort of Chinese children [59,60], to a disorder including epilepsy [61], and to the presence of severe epilepsy in patients with neurodevelopmental disorders [62].

In this study, one new variant, i.e., NC_051816.1 (XM_038684247.1): c.589C > T (p.(Leu197Phe)), was found among the six cases for which the coding DNA of *GRIK2* and the regions surrounding the coding exons were sequenced. All prediction tools estimated that this variant would have a neutral effect on protein function. Therefore, the variant was not examined further. As we only sequenced the coding DNA and splice site regions, the presence of a causal intronic variant cannot be excluded. Furthermore, no brain tissue was available to investigate potential changes in *GRIK2* expression in epileptic dogs. In humans, a downregulation of *GRIK2* has already been shown in patients with epilepsy [63]. Such experiments could provide useful information in the future.

Since the candidate gene within the GWAS region revealed no interesting variants and family-specific variants have been identified in human epilepsy [26], the study cohort was inspected for nuclear families including a case, both seizure-free parents, and a seizure-free sibling to perform WES. Samples were available for two such families, but the DNA concentration for some samples of one family was insufficient. Therefore, WES was performed on the remaining nuclear family to search for family-specific variants in these dogs and as a validation of the GWAS results. Unfortunately, the analysis revealed no variants within the GWAS candidate region. The presence of structural variants was not investigated, as WES is a poor tool to detect these kinds of variants. Future investigations might include high-throughput whole-genome sequencing or long-read sequencing to search for structural variants. Another explanation for the lack of variants in the GWAS candidate region might be the family’s genotype for the risk allele. While the case and mother fit the expected genotype (homozygous and heterozygous for the risk allele, respectively), the healthy father (still seizure-free at 10 years old) and sibling (still seizure-free at 7 years old) were also homozygous for the risk allele. This deviation from the expected phenotype could explain why no variants were found in the candidate region for this family. Nonetheless, WES could still be used as a parallel approach to reveal family-specific variants.

Variants of interest outside of the GWAS candidate region were selected based on gene function, MAF in the EVA database (<10%), and an estimated deleterious effect by at least one prediction tool. After preliminary genotyping of a limited number of samples, the variants NC_051814.1 (XM_038680630.1): c.689C > T (p.(Pro230Leu)) in *CCDC85A* (chromosome 10) and NC_051821.1 (XM_038691182.1): c.493G > A (p.(Glu165Lys)) in *SPAST* (chromosome 17) appeared to be the most promising. According to the ACMG standards and guidelines, these variants were classified as (likely) pathogenic, (likely) benign, or of uncertain significance [37]. Amongst others, we looked at the MAF in population databases. As IE follows a complex inheritance pattern, a (very) low MAF can be expected in these databases. Indeed, the Vt% values were 0.5% and 1.6% for the *CCDC85A* and *SPAST* variants in the EVA database, respectively. Therefore, we accepted these relatively low frequencies as moderate evidence for pathogenicity (PM2).

While little is known about the potential roles of *CCDC85A* and *SPAST* in epilepsy pathogenesis and few functional studies have been conducted, they remain interesting candidate genes. *CCDC85A*’s expression is enhanced in the brain (Human Protein Atlas, proteinatlas.org [64]). Furthermore, *CCDC85A*, *B*, and *C* are delta-interacting protein A (DIPA) family members and potentially participate in N-cadherin-mediated neuronal development through interaction with p120-1, a neuron-specific catenin [65]. N-cadherin regulates the presynaptic function at glutamatergic synapses and controls presynaptic vesicle clustering. It probably has a critical role in maintaining synaptic homeostasis and plasticity [66]. Additional functional studies of *CCDC85A* might further improve the classification of the variant in this study. As for *SPAST*, there is some evidence that it might have a role in the pathogenesis of epilepsy (PS3). This gene encodes spastin, a microtubule-severing enzyme, and the variant is situated in the MT-interacting and trafficking (MIT) domain. It has been established that variants in the MIT domain disrupt lysosomal function [67]. Interestingly, a defective autophagy (in which lysosomes fulfill a vital function) is often associated with neurodevelopmental disorders associated with epilepsy [68]. Furthermore, variants in SPAST are generally known to cause autosomal dominant spastic paraplegia 4 in humans, which is also increasingly associated with additional neurological symptoms such as epilepsy [38]. However, the current knowledge on spastin’s role in epilepsy pathogenesis is insufficient to fulfill ACMG’s PS3 criterion.

Two relevant criteria were fulfilled for the *CCDC85A* variant (one strong (PS4) and one moderate (PM2)), which is sufficient to classify this variant as likely pathogenic. Based on the ACMG guidelines, this variant can be used in human medicine in clinical decision making when combined with other evidence of IE. However, as genetic diversity in dogs tends to be low, we prefer to err on the safe side. Before including this variant, we recommend (1) independent validation of this association, with emphasis on detailed phenotyping, and (2) the collection of family material over multiple generations. Future, additional evidence might change this classification, either in favor or against pathogenicity.

Since only one moderate (PM2) and two supportive criteria are met for the *SPAST* variant (PP2 and PP3), this variant is of uncertain significance according to the ACMG guidelines and should not be used for clinical decision making.

## 5. Conclusions

Although no clear causal variant was found for IE in the DPD, this study is the first to report a new significantly associated SNV for IE on chromosome 12. Further investigations of the *GRIK2* gene or other genes near this locus might reveal more candidate genes for the disease. Furthermore, WES revealed a potentially pathogenic variant in *CCDC85A*. Dogs homozygous for this variant have an increased risk of developing epilepsy. However, we do not recommend the use of either of these two variants as a tool to select against IE before more evidence is gathered.

## Figures and Tables

**Figure 1 animals-13-00810-f001:**
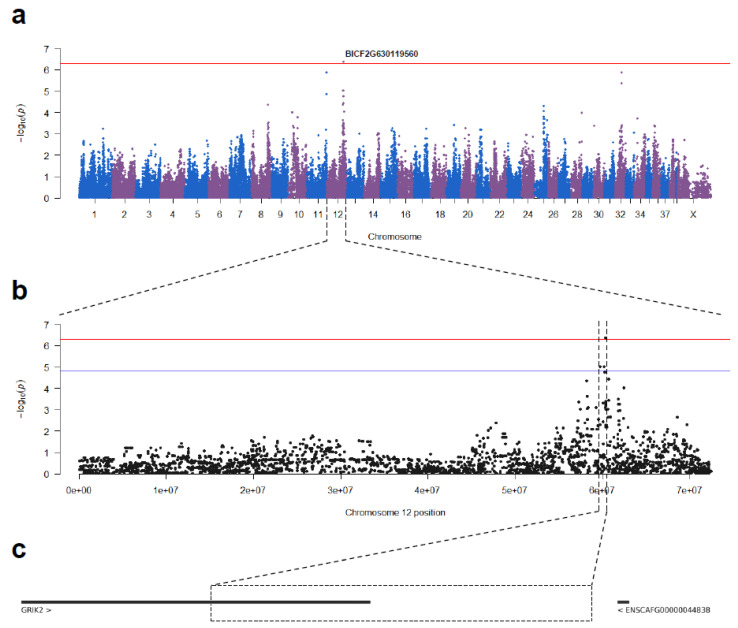
Manhattan plots following univariate linear mixed model testing and genes in the candidate region. (**a**) Manhattan plot with red line indicating the genome-wide threshold. (**b**) Manhattan plot zoomed in on chromosome 12. The red line indicates the genome-wide threshold (threshold = 5.03 × 10^−7^), and the blue line indicates the chromosome-wide threshold (threshold = 1.48 × 10^−5^). (**c**) Genes in the GWAS candidate region according to Ensembl release 104.

**Table 1 animals-13-00810-t001:** Epileptic seizure characterization and medical treatment. Overview of the phenotypical characterization and medical treatment for the selected cases included in this study.

		Absolute	%
Dogs	Total number of dogs	18	-
Sex	Male	8	44%
	Female	10	56%
Age of onset (years)	Mean ± s.d.	3.5 ± 1.4	-
	Median	4	-
	Maximum	5	-
	Minimum	0.6	-
Epileptic seizure frequency	Dogs with a known epileptic seizure frequency	17	-
	≤1 per year	3	18%
	1–2 per year	2	12%
	3–5 per year	2	12%
	6–11 per year	2	12%
	1–2 per month	4	24%
	2–4 per month	3	18%
	≥1 per week	1	6%
Clusters	Occurrence of cluster seizures	7	39%
Prediction	Owner can predict the epileptic seizure	12	67%
Triggers	Epileptic seizures can be provoked	6	33%
	Stress	4	67%
	Sounds	1	17%
	Excitation	2	33%
	Getting startled	1	17%
	Meal	1	17%
	Exercise	0	0%
	Light (flashes/TV/sun)	0	0%
Epileptic seizure duration	<1 min	1	6%
	1–2 min	5	28%
	2–5 min	9	50%
	5–10 min	2	11%
	10–15 min	1	6%
	>15 min	0	0%
Autonomous symptoms	Dog displays at least one autonomous symptom	18	100%
	Salivating	17	94%
	Urinating	13	72%
	Defecation	3	17%
	Loss of consciousness	9	50%
	Loss of consciousness possible/unsure	6	33%
Semiology	Cycling movements	16	89%
	Generalized epileptic seizure evolved from focal	12	67%
	Generalized epileptic seizure	6	33%
	Focal epileptic seizure limited to legs/head	0	0%
	Unknown epileptic seizure type	0	0%
Treatment	Dogs not receiving antiseizure medication	2	11%
	Dogs receiving antiseizure medication	15	83%
	Dogs responding to treatment	11	73%
	Single drug	3	20%
	Phenobarbital	3	100%
	Imepitoin	0	0%
	KBr	0	0%
	Combination	12	80%
	Phenobarbital, KBr	4	33%
	Phenobarbital, imepitoin	1	8%
	Phenobarbital, cannabidiol	2	17%
	Imepitoin, KBr	1	8%
	Phenobarbital, KBr, levetiracetam	1	8%
	Phenobarbital, KBr, cannabidiol	3	25%
	Phenobarbital, KBr, imepitoin, cannabidiol	0	0%

**Table 2 animals-13-00810-t002:** Variants and predicted effects. The variant description is provided at the chromosomal (Chromosome), coding sequence (CDS), and protein (Protein) levels. The rs number (rs) and minor allele frequency in the EVA database (MAF) are provided when available. The outcomes of different online prediction tools are displayed, and deleterious/damaging predictions are indicated in bold.

Variant no.	Chromosome	CDS	Protein	Gene	Rs	MAF	PROVEAN	PolyPhen-2	MutPred2/ Indel
1 *	NC_006589.4: g.38710992_38711006del	XM_022421182.1: c.662_676del	p.(Glu229_Arg233del)	*ENAH*	rs851038082	6.8%	**−4.714**	/	0.271
2	NC_051814.1: g.57941870C > T	XM_038680630.1: c.689C > T	p.(Pro230Leu)	*CCDC85A*	rs852050632	0.5%	**−3.165**	**1**	0.086
3	NC_051814.1: g.64662153A > T	XM_038680741.1: c.51T > A	p.(Asp17Glu)	*VPS54*	/	/	−0.470	**0.993**	0.09
4 **	NC_051816.1: g.60469829C > T	XM_038684247.1: c.589C > T	p.(Leu197Phe)	*GRIK2*	/	/	−1.308	0.201	0.292
5	NC_051821.1: g.25973635G > A	XM_038691182.1: c.493G > A	p.(Glu165Lys)	*SPAST*	rs850566951	1.6%	**−3.165**	**0.997**	**0.643**
6	NC_051842.1: g.8412424G > C	XM_038448399.1: c.578G > C	p.(Arg193Pro)	*BRINP3*	rs852865827	0.7%	**−4.929**	**0.999**	**0.813**

* Variant cannot be mapped to ROS_Cfam_1.0 and is displayed here according to CanFam3.1. ** Variant found in the GWAS region of interest.

**Table 3 animals-13-00810-t003:** Genotyping results for the *CCDC85A* and *SPAST* variants. The number of homozygous wild-type (Wt/Wt), heterozygous (Wt/Vt), and homozygous variant (Vt/Vt) dogs, as well as the variant allele frequency (Vt%), is shown for the case group (Cases), the control group (Controls), and the total population (Tot_pop).

	*CCDC85A*				*SPAST*			
	Wt/Wt	Wt/Vt	Vt/Vt	Vt%	Wt/Wt	Wt/Vt	Vt/Vt	Vt%
Cases	11	2	5	33.3%	11	6	1	22.2%
Controls	67	27	6	19.5%	82	16	2	10.0%
Tot_pop	186	82	24	22.3%	242	47	3	9.1%

**Table 4 animals-13-00810-t004:** Variant classification of the XM_038680630.1: c.689C > T variant. Criterion numbers are shown as provided by the American College of Medical Genetics. Results, remarks, and conclusions for each criterion are listed.

Criterion	Result	Remarks	Conclusion
Significant OR > 5 (PS4)	OR: 6.0; 95% CI: 1.6–22.6		PS4 fulfilled
Low MAF in population databases (PM2)	0.5% in EVA database	Homologous variant in humans has an much lower frequency in multiple population studies	PM2 fulfilled
Multiple lines of computational evidence (PP3/BP3)	2/4 programs predicted a deleterious effect, and 2/4 predicted a neutral effect.	When in silico predictions disagree, this evidence should not be used to classify a variant	PP3/BP3 cannot be used
Variant segregation (PP1/BS4)	N/A	Insufficient family material available to investigate cosegregation in multiple affected family members	PP1/BS4 cannot be used
Functional studies show deleterious (PS3) or no deleterious (BS3) effect	Enhanced expression in the brain; possible interaction with neuron-specific catenin p120-1, which probably plays a critical role in synaptic homeostasis and plasticity	Too few functional studies exist to fulfill these criteria	PS3/PS3 not fulfilled

## Data Availability

The data generated during this study can be found within the published article and its supporting information. The variant data for this study have been deposited in the European Variation Archive (EVA) at EMBL-EBI under accession number PRJEB56315. The raw WES data have been submitted to the Sequence Read Archive (SRA) under accession number PRJNA891496, samples SAMN31329656—SAMN31329659.

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
