# Peer review of "Identification of a Novel Idiopathic Epilepsy Risk Locus and a Variant in the *CCDC85A* Gene in the Dutch Partridge Dog"

_animals, 2023, doi:10.3390/ani13050810_

Round 1

Reviewer 1 Report

The paper provides a potential novel risk loci for idiopathic epilepsy in the Dutch partridge dog using both a GWA analysis and whole-exome sequencing. The authors have made substantial effort to advance the knowledge of genetic risk factors, while identifying the limitations of the research approach.

One of the strengths of this paper is the two-pronged approach to identifying variants while capitalizing on information from pet owners, as well as using a group of closely related dogs to increase the likelihood of detecting risk variants.

The authors have also done a good job of discussing the limitations of current genetic studies on idiopathic epilepsy in dogs, and highlighting the potential challenges facing future research in this field.

Overall, this paper contributeds to the field of canine genetics and will be of interest to veterinarians, breeders, and pet owners alike.

Author Response

Dear reviewer,

Thank you for your time and effort in reading and reviewing this paper.

Best wishes,

Evy Beckers

Reviewer 2 Report

This manuscript presents a novel region of interest and a potential variant related to epilepsy in a specific dog breed. The design, analyses, and overall conclusions are sound. However, the data have to be made publically available as of now I was not able to find the stated accession numbers. Furthermore, improvements to the Methods description and shortening of Discussion while including the relevant information in Results should be done. I list the specific comments below.
Abstract: Please, state chromosome number for the CCDC85A variant. Introduction: Should it be 4 variants? What about the PITRM1-related epilepsy (https://www.omia.org/OMIA002324/9615/)? Lafora disease has been reported in many different breeds, not just the initial dachshund. The correct symbol for DIRAS family GTPase 1 is "DIRAS1". I suggest adding OMIA numbers to the epilepsy-associated genes to make it clearer and easier to find all the affected breeds and additional details. M&M: Overall it lacks specifics which makes it difficult to understand the individual steps. How many dogs cases vs controls were recruited/selected/array genotyped? Only the WES part is clear on that. How many markers were left after the 4-step QC? How many markers were used for PCA? How many variants were detected and genotyped in the selected genes? 2.2: It is introduced before as "genome-wide association study (GWAS)", stay consistent throughout the manuscript. Please, add basic information on how Bonferroni and ORs calculations were made, everything else is described in great detail, so this part feels like something is missing. Starting from "The GRIK2 gene was an interesting candidate...", it should be a separate subsection for better readability. Primer BLAST should have its own reference: https://pubmed.ncbi.nlm.nih.gov/22708584/ 2.3.: "While GWAS was used to detect common genes,..." - I would think it was used to detect IE-associated regions? Why were the exoms not aligned to the better quality assembly, as is mentioned for the GWAS data (ROS_Cfam_1.0)? How many and of how many breeds are there in the variant database? Are there any DPD included? ! I do not see this variant database: "The following term was not found in SRA: PRJNA891496." 2.4.: Candidate variants were only identified by WES, not GWAS. GWAS identifies a region of interest, in which you found a positional candidate gene that you subsequently sequenced. How many variants were detected this way in GRIK2? Results 3.1.: What does "some closely related individuals" mean? Do you mean 1st degree relatedness? How many siblings or parent-offspring pairs were there? Were these in cases or controls or both? 3.1.2.: This part is really confusing. Only 24 out of 45 dogs were diagnosed in detail? And from those only 5 actually received the IE diagnosis? 3.1.3.: How did the other 3 enter the study if they didn't fill the questionnaire? Were these 18 part of the 24 described in 3.1.2.? How many of the final cases were males and females? 3.2.: How big was the region (lines 260-263)? Figures should be named sequentially. Therefore, S1 needs to be the manhattan and S2 the PCA plot. Line 273 - "The 0.5 Mb associated region..." Was that same/smaller/larger than the original region? Were there more/less/same number of genes? Line 280 "variant allele frequency is reported in the EVA database (47% and 32% for rs8745274 and rs850859789, respectively)" - can you provide exact link where these numbers come from? Because I see only MAF=0.435 for rs8745274 and MAF=0.074 for rs850859789, when searching those positions. Table 2 description should read "bold". "** variant found with GWAS." should read "** variant found in the GWAS region of interest." 3.3.: I believe the last sentence should point to Table 2. 3.4.: Same as above. There is no ACMG classification as mentioned in the title. Discussion: Have you looked at potential structural variants around the GWAS region in the sequenced dogs? What does "PP1/BS4" mean? Please, define all abbreviations accordingly. Table 4 and most of the text between lines 458 and 507 should be presented in Results. This section needs to be split and relevant results included as mentioned in the title of 3.4. while the discussion parts should stay in chapter 4. This will appropriately complete the Results and shorten the very long Discussion.

Author Response

Dear reviewer,

Thank you for the time and effort put into thoroughly reviewing this paper. Please find the response to your comments below.

Kind regards,

Evy Beckers

This manuscript presents a novel region of interest and a potential variant related to epilepsy in a specific dog breed. The design, analyses, and overall conclusions are sound. However, the data have to be made publically available as of now I was not able to find the stated accession numbers.

  • Upon submission of the data, we requested them to remain private until the article was accepted. We have sent a request to make them publicly available.

Furthermore, improvements to the Methods description and shortening of Discussion while including the relevant information in Results should be done. I list the specific comments below.
Abstract: Please, state chromosome number for the CCDC85A variant.

  • The chromosome number was added (lines 37-38)

Introduction: Should it be 4 variants? What about the PITRM1-related epilepsy (https://www.omia.org/OMIA002324/9615/)?

  • We have added this 4th variant to lines 65-67.

Lafora disease has been reported in many different breeds, not just the initial dachshund.

  • We specified it was first reported in this breed (line 59)

The correct symbol for DIRAS family GTPase 1 is "DIRAS1".

  • This was adapted accordingly (line 62)

I suggest adding OMIA numbers to the epilepsy-associated genes to make it clearer and easier to find all the affected breeds and additional details.

  • The OMIA numbers were added (lines 60, 61, 63 and 66)

M&M: Overall it lacks specifics which makes it difficult to understand the individual steps. How many dogs cases vs controls were recruited/selected/array genotyped? Only the WES part is clear on that. How many markers were left after the 4-step QC? How many markers were used for PCA? How many variants were detected and genotyped in the selected genes?

  • We omitted this information in the M&M section since we thought it was more appropriate for the results section. To improve the clarity of the paper, we now also added this information in the M&M section.
  • Line 88 now mentions the total number of dogs that entered the study, line 96 the number of dogs with epilepsy (according to the owner), and line 114 the number of cases/controls that were genotyped on the SNV array. The remaining number of SNVs after QC is now mentioned on line 125 and those for the PCA on line 129.
  • The number of variants called in the WES was added on line 166. The number of coding, non-synonymous variants was added on line 174 and the number remaining in the end on line 180.

2.2: It is introduced before as "genome-wide association study (GWAS)", stay consistent throughout the manuscript.

  • GWA was changed to GWAS on lines 115, 125, 223, 265, 437, and 545 as well as in the supplementary file.

Please, add basic information on how Bonferroni and ORs calculations were made, everything else is described in great detail, so this part feels like something is missing.

  • A reference and a brief explanations was added to explain the Bonferroni correction (line 134).
  • A supplemental table (Table S1) was added explaining the two different OR calculations used in the manuscript. References to the table were added to lines 136, 193, and 205).

Starting from "The GRIK2 gene was an interesting candidate...", it should be a separate subsection for better readability.

  • A new subsection “GWAS candidate gene” was introduced (line 138)

Primer BLAST should have its own reference: https://pubmed.ncbi.nlm.nih.gov/22708584/

  • Thank you for this reference. It was added on line 145.

2.3.: "While GWAS was used to detect common genes,..." - I would think it was used to detect IE-associated regions?

  • Indeed, by detecting IE-associated regions, common genes are also (though more indirectly) detected (here, the GRIK2 gene). To avoid confusion, we rephrased this sentence (line 154).

Why were the exoms not aligned to the better quality assembly, as is mentioned for the GWAS data (ROS_Cfam_1.0)?

  • As the exome-plus design (which was used for the WES in this article) was based on CanFam3.1, we also performed the alignment on this assembly.

How many and of how many breeds are there in the variant database? Are there any DPD included? ! I do not see this variant database: "The following term was not found in SRA: PRJNA891496."

  • As mentioned above, we have asked to make the submission public and you should have access to it soon. No DPD are included in the database that was used for filtering. The breeds included are Labrador retriever (16), deerhound (7), Belgian shepherd (5), bloodhound (5), crossbreeds (5), English cocker spaniel (1), pitbull (1), Irish setter (1), American Staffordshire (1), Chesapeake Bay retriever (1), French bulldog (1), and Jack Russell terrier (1).

"Regarding the two databases we submitted our data to, the data is already public in one. The EVA database just e-mailed me to let me know the data will be made public over the weekend. (on 2/17 2023)

2.4.: Candidate variants were only identified by WES, not GWAS. GWAS identifies a region of interest, in which you found a positional candidate gene that you subsequently sequenced. How many variants were detected this way in GRIK2?

  • We added a sentence at the end of 2.3 disclosing how many variants were found (line 151). The first sentence of 2.5 was rephrased (line 181).

Results 3.1.: What does "some closely related individuals" mean? Do you mean 1st degree relatedness? How many siblings or parent-offspring pairs were there? Were these in cases or controls or both?

  • The closely related individuals include first-degree relatives (parents, siblings), but also a few second-degree relatives. For most of the dogs with epilepsy, one or more samples of epilepsy-free siblings were available. Two or more full siblings are available for 52 litters. For 25 litters, the sire was available (14 sires in total) and for 32 litters, the dam was available (22 dams in total). The relatedness between all the dogs of the study sample proved too complex (several popular sires and dams, full and half-siblings, the availability of one or both parents, cousins, …) to explain very clearly. However, we did include the above information in lines 214-217.

This relatedness could be the cause of the population stratification we saw in the GWAS. However, this was corrected.

3.1.2.: This part is really confusing. Only 24 out of 45 dogs were diagnosed in detail? And from those only 5 actually received the IE diagnosis?

  • Indeed, only 24 of the dogs had a blood examination done by their veterinarian and we had access to the results for 10 of those dogs. 5 dogs got a Tier I or II diagnosis, which is a confidence level that is described by the IVETF. As the diagnosis is one of exclusion, more confidence is gained as more evidence is gathered. This does not mean that none of the other 40 dogs were diagnosed with IE, only that they did not reach the Tier I criteria.

3.1.3.: How did the other 3 enter the study if they didn't fill the questionnaire?

  • The owners of these 3 dogs sent in a blood sample, but never answered the questionnaire. These dogs were therefore not included as a case nor as a control. They were used to estimate the allele frequencies of the population.

Were these 18 part of the 24 described in 3.1.2.?

  • Not all of the 18 cases received a blood examination. As explained in the M&M (2.1.2.), inclusion as a case was based on strict criteria extracted from the questionnaire. We did not require a Tier I level of confidence.

How many of the final cases were males and females?

  • 10 were female and 8 were male. This information was added to Table 1.

3.2.: How big was the region (lines 260-263)?

Line 273 - "The 0.5 Mb associated region..." Was that same/smaller/larger than the original region? Were there more/less/same number of genes?

  • The region without correction for population stratification was bigger and included more genes. However, as these are biased results (because of population stratification), they are untrustworthy. Therefore, the authors feel it is redundant to describe how big this region was and how many/which genes this region harbored.

Figures should be named sequentially. Therefore, S1 needs to be the manhattan and S2 the PCA plot.

  • The Manhattan plot is now named S1, the QQ plots S2 (as they are mentioned before the PCA), and the PCA S3. The numbering was also adjusted in the text (lines 271-274).

Line 280 "variant allele frequency is reported in the EVA database (47% and 32% for rs8745274 and rs850859789, respectively)" - can you provide exact link where these numbers come from? Because I see only MAF=0.435 for rs8745274 and MAF=0.074 for rs850859789, when searching those positions.

  • rs8745274: T/C variant, location 12:59594951 on CanFam3.1 (other accessions were not available at the time of writing)
  • rs850859789: The A/C variant, location 12: 59636168 on CanFam3.1
  • For both variants, an allele frequency is mentioned under Population Statistics (0.435 and 0.311, respectively). However, these only include Run 2. Under Genotypes, the genotypes for all samples can be viewed for Run 1 and Run 2. The allele frequency was calculated based on this information.

"Regarding the two databases we submitted our data to, the data is already public in one. The EVA database just e-mailed me to let me know the data will be made public over the weekend. (on 2/17 2023)

Table 2 description should read "bold". "** variant found with GWAS." should read "** variant found in the GWAS region of interest."

  • This was adapted (lines 303 and 305)

3.3.: I believe the last sentence should point to Table 2.

  • This was adapted (line 317)

3.4.: Same as above. There is no ACMG classification as mentioned in the title.

Table 4 and most of the text between lines 458 and 507 should be presented in Results. This section needs to be split and relevant results included as mentioned in the title of 3.4. while the discussion parts should stay in chapter 4. This will appropriately complete the Results and shorten the very long Discussion.

  • Parts of the discussion were added to the results section, as suggested.

Discussion: Have you looked at potential structural variants around the GWAS region in the sequenced dogs?

  • As WES is not an ideal tool to detect structural variants (they are very difficult to detect, more so than with other techniques like long-read sequencing or WGS), we did not look for these variants. We have added some discussion about this subject in lines 477-480.

What does "PP1/BS4" mean? Please, define all abbreviations accordingly.

  • These are codes used by the ACMG rather than abbreviations. We endeavored to make it more clear what the codes mean in lines 333-334.